# Synthesis and Molecular Docking Studies of Alkoxy- and Imidazole-Substituted Xanthones as α-Amylase and α-Glucosidase Inhibitors

**DOI:** 10.3390/molecules28104180

**Published:** 2023-05-18

**Authors:** Dolores G. Aguila-Muñoz, Gabriel Vázquez-Lira, Erika Sarmiento-Tlale, María C. Cruz-López, Fabiola E. Jiménez-Montejo, Víctor E. López y López, Carlos H. Escalante, Dulce Andrade-Pavón, Omar Gómez-García, Joaquín Tamariz, Aarón Mendieta-Moctezuma

**Affiliations:** 1Centro de Investigación en Biotecnología Aplicada, Instituto Politécnico Nacional, Carretera Estatal Santa Inés Tecuexcomax-Tepetitla, Km 1.5, Tepetitla de Lardizábal, Tlaxcala 90700, Mexico; lupaita@gmail.com (D.G.A.-M.); eagle_vl@hotmail.com (G.V.-L.); erika16.st@gmail.com (E.S.-T.); ccruzl@ipn.mx (M.C.C.-L.); fejimenezm@ipn.mx (F.E.J.-M.); vlopezyl@ipn.mx (V.E.L.y.L.); 2Departamento de Química Orgánica, Escuela Nacional de Ciencias Biológicas, Instituto Politécnico Nacional, Prol. Carpio y Plan de Ayala S/N, Mexico City 11340, Mexico; escalantecah@gmail.com (C.H.E.); gogamanj@hotmail.com (O.G.-G.); jtamariz@yahoo.com.mx (J.T.); 3Departamento de Fisiología, Escuela Nacional de Ciencias Biológicas, Instituto Politécnico Nacional, Av. Wilfrido Massieu S/N, Mexico City 11340, Mexico; andrade_eclud88@hotmail.com; 4Departamento de Microbiología, Escuela Nacional de Ciencias Biológicas, Instituto Politécnico Nacional, Prol. Carpio y Plan de Ayala S/N, Mexico City 11340, Mexico

**Keywords:** diabetes mellitus, α-glucosidase, α-amylase, alkoxy-substituted xanthones, imidazole-substituted xanthones

## Abstract

Current antidiabetic drugs have severe side effects, which may be minimized by new selective molecules that strongly inhibit α-glucosidase and weakly inhibit α-amylase. We have synthesized novel alkoxy-substituted xanthones and imidazole-substituted xanthones and have evaluated them for their in silico and in vitro α-glucosidase and α-amylase inhibition activity. Compounds **6c, 6e**, and **9b** promoted higher α-glucosidase inhibition (IC_50_ = 16.0, 12.8, and 4.0 µM, respectively) and lower α-amylase inhibition (IC_50_ = 76.7, 68.1, and >200 µM, respectively) compared to acarbose (IC_50_ = 306.7 µM for α-glucosidase and 20.0 µM for α-amylase). Contrarily, derivatives **10c** and **10f** showed higher α-amylase inhibition (IC_50_ = 5.4 and 8.7 µM, respectively) and lower α-glucosidase inhibition (IC_50_ = 232.7 and 145.2 µM, respectively). According to the structure–activity relationship, attaching 4-bromobutoxy or 4′-chlorophenylacetophenone moieties to the 2-hydroxy group of xanthone provides higher α-glucosidase inhibition and lower α-amylase inhibition. In silico studies suggest that these scaffolds are key in the activity and interaction of xanthone derivatives. Enzymatic kinetics studies showed that **6c**, **9b**, and **10c** are mainly mixed inhibitors on α-glucosidase and α-amylase. In addition, drug prediction and ADMET studies support that compounds **6c**, **9b**, and **10c** are candidates with antidiabetic potential.

## 1. Introduction

Diabetes mellitus is a metabolic disorder with increasing prevalence. This disease now afflicts about 537 million people worldwide according to the International Diabetes Federation, and thus represents a public health problem [1]. It is characterized by hyperglycemia stemming from congenital or acquired insulin secretion deficiency. The best therapeutic approach known to date consists of inhibiting intestinal enzymes responsible for carbohydrate hydrolysis, such as α-amylase and α glucosidase. Of the commercially available antidiabetic drugs, α-glucosidase inhibitors capable of acting in the intestine seem to be most effective for reducing postprandial hyperglycemia [2].

α-Amylase is an enzyme (EC. 3.2.1.1) secreted by the pancreas and salivary glands that hydrolyzes α-linked polysaccharides (e.g., starch and glycogen) to maltose [3]. α-Glucosidase (EC. 3.2.1.20), an enzyme found in the small intestine, catalyzes the cleavage of the α-1,4-glycosidic bonds of maltose to form glucose [4]. Currently, antidiabetic α-amylase and α-glucosidase inhibitors (e.g., acarbose, miglitol, and voglibose) are based on carbohydrate-related structures and are effective. However, their use is limited by the adverse effects of flatulence, abdominal pain, and diarrhea, which could result from the fermentation of undigested carbohydrates derived from the strong inhibition of α-amylase [5,6,7]. Therefore, it is desirable to design new selective molecules with strong inhibition of α-glucosidase and weak inhibition of α-amylase to minimize side effects.

Several heterocyclic compounds containing oxygen and nitrogen are relevant for designing and developing new drugs. For instance, xanthone is a dibenzo-γ-pyrone heterocycle that has drawn the attention of researchers due to its broad spectrum of biological activity [8,9,10,11]. Xanthone derivatives are oxygenated heterocyclic compounds that occur as secondary metabolites in some families of higher plants (Guttiferae, Gentianaceae, Moraceae, Clusiaceae, and Polygalaceae). Prenylated xanthones such as α-mangostin (Figure 1) exhibit a broad spectrum of biological activity, such as antimicrobial [12,13], antioxidant [14], antiviral [8], and anticancer properties [15,16], and also express multiple target proteins [17,18,19,20,21]. The functionalization of α-mangostin has led to antimicrobial and anticancer agents with significantly improved pharmacological properties [16,22,23,24,25].

The functionalization of xanthones as α-glucosidase inhibitors has revealed the key factors involved in their observed inhibitory activity: the formation of an H-bond, hydrophobic groups with a π-conjugated system, and flexibility in conformation [21,26]. This is the case of 2-hydroxy-3-methoxyxanthone **1** (Figure 1), a natural product found in plants of the genus *Hypericum* (Clusiaceae), which exhibits anticancer properties [27,28,29]. Thus, it would be interesting to use this compound as a molecular platform to generate new derivatives with potential pharmaceutical applications. 

On the other hand, imidazole-containing heterocycles exhibit anticancer [10,30], antimicrobial [31,32,33], anti-inflammatory [34], and antidiabetic properties [35,36,37,38,39,40]. The functionalization of the imidazole scaffold with aryl substituents has significantly improved the inhibitory effect on α-glucosidase [37,38,41,42,43,44]. Hence, compounds designed with an imidazole ring on the xanthone framework may exert an important inhibitory effect on α-amylase and α-glucosidase (Figure 1). 

Based on the aforementioned observations, the aim of the current contribution was to synthesize novel alkoxy-substituted xanthone derivatives and imidazole-substituted xanthones and test their potential as α-amylase and α-glucosidase inhibitors. Three structural elements were considered presently: (i) a natural xanthone core as an antidiabetic pharmacophore fragment, (ii) alkoxy groups substituted with a π-system or imidazolyl rings with drug-like properties, and (iii) a medium-chain alkoxy group to enhance lipophilicity. Compounds containing these elements were evaluated for their in vitro activity as α-amylase and α-glucosidase inhibitors. Moreover, in silico studies were performed to gain insight into the interaction of the compounds with the active site of α-amylase and α-glucosidase enzymes.

## 2. Results and Discussion

### 2.1. Chemistry

The synthetic pathways of alkoxy-substituted xanthone derivatives **5**, **6a–d**, **7**, and **8** (Figure 1) started from the acylation of phenol **2** with 2-iodobenzoyl chloride in the presence of boron trifluoride diethyl etherate, thus obtaining benzophenone **3a**. The latter compound was *O*-alkylated by a reaction with methyl 2-bromoacetate (**4a**) to afford **3b**, which was treated with *N*,*N*-dimethylformamide dimethyl acetal (DMFDMA) (4.0 mol equiv.) at 120 °C to provide the corresponding xanthonyl enaminone **5** and xanthone **6a** in low to moderate yields. This pathway involves a cascade reaction that possibly occurs through intramolecular cyclization followed by condensation with DMFDMA. Xanthone **1** was obtained by the cyclization of the benzophenone intermediate **3a** with a solution of KOH in water at 100 °C for 6 h [45]. Alkylation of compound **1** with α-halocarbonyls **4a–c** or allyl bromide (**4d**) in the presence of K_2_CO_3_ produced alkoxy-substituted xanthones **6a–d** in high yields (Figure 1). The oxyallyl xanthone **6d** was subjected to a Claisen rearrangement to furnish allylhydroxy-substituted xanthone **7** in excellent yield. 2-Hydroxyxanthone **1** was also reacted with 1,1-diethoxy-3-methylbut-2-ene in the presence of 3-methylpicoline, leading to pyranoxanthone **8** in modest yield.

The two series of imidazole-substituted xanthones **10a–f** and **12a–f** were prepared by attaching imidazoles to the xanthones **9a–b** and **11a–b**, respectively. Firstly, alkylation of the phenoxy group of xanthones **1** and **7** with 1,4-dibromobutane under base conditions generated xanthones **9a–b** in excellent yields. Substitution of the bromine atom by the 2-substituted imidazoles **13a–c** resulted in the corresponding imidazole-substituted xanthones **10a–f** in moderate to good yields [22].

For **12a–b**, the intermediate epoxides **11a–b** were formed by alkylation of **1** and **7** with epichlorohydrin in ethanol in the presence of KOH. The epoxide ring of derivatives **11a–b** was opened with imidazoles **13a–c** in methanol to convert them into the respective compounds **12a–f** in moderate to good yields [25]. The alkylation of **7** with α-halocarbonyl **4c** delivered alkoxy-substituted xanthone **6e** in excellent yield (Figure 2).

All the synthesized compounds were fully characterized by ^1^H-NMR, ^13^C-NMR, and HRMS. In the ^1^H-NMR spectra, for compounds **6a–e**, the methylene protons of the acetophenone moiety appeared as a singlet at 4.74–5.74 ppm. For compounds **10a–f**, the methylene protons adjacent to the nitrogen atom of the butoxyimidazole moiety were observed as triplets at 4.03–4.22 ppm, while compounds **12a–f** were observed as a doublet of doublets at 4.06–4.22 ppm. All the peak values from 6.84–8.32 ppm were assigned to aromatic protons. In the ^13^C-NMR spectra, the carbonyl groups of xanthone and acetophenone appeared at 174.5–175.9 and 193.3–203.7 ppm, respectively.

In the case of xanthonyl enaminone **5**, a single stereoisomer was obtained, and its Z geometry was established by NOE experiments. Irradiation of the signal assigned to the methyl protons of the dimethylamine group produced the enhancement of the signal corresponding to the aryloxy ring of the xanthone scaffold. This stereoselectivity has been observed in similar systems, probably because of the greater stability of the planar π-conjugated acrylate system when the bulky dimethylamine group is located at the opposite side of the double bond [46].

### 2.2. In Vitro α-Glucosidase Inhibition

After testing each compound for its inhibitory effect on α-glucosidase, this result was compared to the effect of acarbose (**14**), an antidiabetic drug known to inhibit the α-glucosidase and α-amylase enzymes [6]. Initially, the compounds were evaluated at 400 µM. Structurally, alkoxy-substituted xanthones were divided into two groups based on the nature of the alkoxy chain substituents at the C-1 and/or C-2 positions of the xanthone core: (1) substituted alkoxy derivatives **5**, **6a–e**, **8**, **9a–b**, and **11a–b** and (2) substituted imidazolyl derivatives **10a–f** and **12a–f**. Xanthone **1** at 400 µM exhibited weak inhibitory activity on α-glucosidase (Table 1), as did **6a** (with a 2-oxyacetate substituent at the 2-hydroxy group). An insertion of the (4-chlorophenyl)-2-oxoethoxy substituent afforded **6c**, increasing the inhibitory effect (IC_50_ = 143.6 ± 0.17 µM for **6a** and IC_50_ = 16.0 ± 0.03 µM for **6c**). Contrarily, the presence of an α-acetonyl or allyl group (**6b** and **6d**) resulted in a weak inhibition effect. Similarly, a significant decrease in inhibitory activity was found with **5**, formed by the attachment of an enaminone moiety to the 2-oxyacetate group of **6a**. 

Interestingly, **7** (with an allyl group at the C-1 position) generated a greater inhibitory effect on α-glucosidase (IC_50_ = 196.4 ± 0.07 µM) than its analogs **1** and **8**. The alkoxy-substituted xanthone derivatives **6e**, **9a–b**, and **11a–b** were examined to clarify the role of the allyl group, which in **9b** provided the most potent inhibitory activity (IC_50_ = 4.00 ± 0.007 µM). The introduction of an imidazole ring instead of the bromine group at the C-4 position of the chain in **9b** furnished compounds **10a–d** and **10f**, resulting in a significant decrease in the inhibitory effect. Regarding the imidazolyl-substituted xanthones series **12a–f**, **12b** and **12e** (with a phenyl group in the 2-imidazole ring) showed higher inhibitory activity (IC_50_ = 112.8 ± 0.12 µM and 104.9 ± 0.01 µM, respectively) than **12a** and **12d**. According to the results, the introduction of the imidazolyl group had no significant effect on activity since almost all the products (**10a–d**, **10f**, and **12a–e**) are less effective than their intermediates (**9a–b** and **10a–b**).

In summary, the inhibition of α-glucosidase produced by acarbose (**14**, the reference drug) was significantly improved when a phenyl or aryl ring was present at the C-2 side chain combined with the 2-substituted imidazole ring of the xanthone core, leading to **6c**, **6e**, **9a–b**, **12b**, and **12f**. Considering structural similarities among other molecules, inhibition was much stronger for those containing the C-1 allyl group (**7**, **9b**, and **11b**). The data suggest that a molecule with an allyl or substituted aryl group has better affinity for the amino acid residues of the α-glucosidase enzyme and thus enhanced bioactivity, which owes itself to π-sticking or hydrophobic effects [11,21,37], as well as interactions with the halogen atoms substituted at the phenyl groups or at the alkyl side chain [36,38,47]. 

### 2.3. In Vitro α-Amylase Inhibition

All compounds evaluated on α-glucosidase were tested for their capacity to diminish α-amylase activity (Table 1). The compounds were evaluated at 100 µM. Natural xanthone **1** without any substituent modification at the C-2 position exerted very strong inhibition of α-amylase. The incorporation of a 2-oxyacetate group at C-2 in compound **6a** did not show activity. A limited inhibitory effect was observed with the insertion of a (4-chlorophenyl)-2-oxoethoxy substituent in compound **6c** or a 4-bromobutoxy substituent in compound **9a**. The IC_50_ value was higher for **6c** and **9a** than for acarbose (**14**), indicating a lower inhibitory effect for these two compounds. No inhibitory activity was detected for **7** with an allyl group at the C-1 position, while a small inhibitory effect was found for **6e**, which was prepared by introducing a (4-chlorophenyl)-2-oxoethoxy substituent at C-2. There was limited inhibitory activity with **9a** and sharply increased activity when the bromine atom at the side chain of this analogue was replaced by a 2-(4-chlorophenyl)imidazol-1-yl group to provide **10c**.

This increase also occurred for **10f**. On the other hand, the introduction of an additional OH group at the alkoxy side chain of compounds **12a–f** led to an absence of inhibitory activity.

In summary, greater inhibition of α-glucosidase and lesser inhibition of α-amylase were achieved by the incorporation of a (4-chlorophenyl)-2-oxoethoxy moiety (**6c** and **6e**) and 4-bromobutoxy (**9b**) at C-2 of the xanthone scaffold. In contrast, the lowest inhibition of α-glucosidase and highest inhibition of α-amylase were found with natural xanthone **1** and with the addition of the 4-(2-(4-chlorophenyl)imidazol-1-yl)butoxy moiety (i.e., **10c** and **10f**). Considering the aforementioned desirability of a low inhibitory effect on α-amylase (to avoid gastrointestinal side effects) together with potent inhibitory activity on α-glucosidase, compounds **6c**, **6e,** and **9b** are promising candidates for the development of antidiabetic drugs.

### 2.4. Enzymatic Kinetic Study

In order to explore the interaction mechanism of alkoxy-xanthones **6c**, **9b**, and **10c**, the type of inhibition exhibited by these selective inhibitors was analyzed using Lineweaver–Burk plots (double reciprocal). The *X*-axis values represent the reciprocal for the α-glucosidase substrate, *p*-nitrophenyl-α-_D_-glucopyranoside (*p*-NPG), while for the α-amylase substrate they represent starch, thus being 1/(starch). The *Y*-axis values are the reciprocal of the reaction velocity (Vo), thus being 1/Vo. Given that the plots did not intersect the *X*- or *Y*-axis, the inhibition of α-glucosidase exerted by these compounds is carried out in mixed mode (Figure 2A). The K_I_ values of **6c**, **9b**, and **10c** are 21.5, 1.25, and 139.0 µM, respectively. The K_I_ values for these compounds are less than K_m_, indicating that they have a higher affinity for the enzyme than the substrate used in the assay [48].

The amylase plots made it possible to determine that compound **6c** is a competitive type of inhibitor, while **10c** is a mixed-type inhibitor. The K_I_ values of **6c** and **10c** are 63.2 and 2.2 µM, respectively (Figure 2B). These K_I_ values indicate that **6c** has greater affinity for the enzyme.

### 2.5. Evaluation of Antioxidant Activity

Since antioxidants contribute to the prevention of diabetes mellitus and other diseases [49], the antioxidant potential of the synthesized compounds was determined. The capacity for free radical scavenging was assessed by means of the 2,2-diphenyl-1-picrylhydrazyl (DPPH) radical method, with butylhydroxytoluene (BHT) as the positive control. A decrease in color intensity represents the scavenging of DPPH (Table 1), which was calculated as a percentage. The alkoxy-substituted xanthones **6a–e**, **9a–b**, and **11a–b**, as well as imidazole-substituted xanthones **10a–f** and **12a–f,** did not show any significant antiradical activity even at the maximum concentration tested. Only compounds **1** and **7** were able to scavenge DPPH to some extent (ca. 33% and 46% at 2.5 mM), suggesting that this effect is mainly related to the hydroxy group substituted at the C-2 position. 

### 2.6. Molecular Docking Analysis

To explore the binding interactions of the most active compounds, molecular docking studies were carried out between alkoxy-substituted xanthones **6c**, **6e**, **9b**, and **10c** and the isomaltase enzyme (the α-glucosidase of *S. cerevisiae*), as well as **1**, **6c**, **9a**, **10c**, and **10f** and the human α-amylase enzyme. The results of the interactions are illustrated in 2D and 3D (Figure 3 and Figure 4), revealing that the alkoxy-substituted xanthone derivatives recognized some of the key amino acid residues in the catalytic pocket, such as His112, Arg213, Asp215, Glu277, His351, Asp352, and Arg442. A similar set of residues is reported by maltose inhibitors [50,51,52,53,54].

Regarding the binding energy of the compounds with the enzymes, the alkoxy-substituted xanthone derivatives (**6c**, −9.17; **6e**, −9.41; **9b**, −7.89 kcal/mol) and the imidazole-substituted xanthone analogues (**10c**, −9.82; **10f**, −9.66 kcal/mol) (Table 2 and Table 3) have better binding energy values (Δ*G*) than the reference drug **14** (−7.78 for α-glucosidase and −2.92 for α-amylase). The docking studies with α-glucosidase reveal that compounds **6c** and **6e** adopt an S-shaped conformation. Moreover, the dibenzo-γ-pyrone system of xanthone is involved in hydrophobic interactions of various types: π-π stacked (Phe303), π-π T-shaped (Tyr158), π-anion (Glu411, Asp352), and π-cation and π-sigma (Arg315). The (4-chlorophenyl)-2-oxoethoxy fragment at C-2 of the xanthone core displays different hydrophobic interactions, including π-π T-shaped (Tyr72), π-alkyl (His112, Val216, and Phe178), and π-anion (Asp215 and Glu277), as well as hydrogen bond interactions with Gln279 and Asp352. For compound **9b**, the allyl group at C-1 shows a π-alkyl hydrophobic interaction with Phe303, and the 4-bromobutoxy fragment at C-2 exhibits an alkyl hydrophobic interaction with Arg315 and a hydrophilic interaction with the halogen atom (Thr310) while forming a carbon–hydrogen bond with Glu411 and Arg315 residues. According to these results, **6c**, **6e**, and **9b** bind to an allosteric site near the catalytic site of the enzyme since they docked with amino acids of the catalytic pocket, thus sterically blocking it and indicating that they act as mixed inhibitors [55]. Data of **10c** are summarized in Appendix A.

The same trend of interactions was observed when docking natural xanthone **1** and acarbose (**14**) with α-amylase, exhibiting interactions with some amino acid residues at the active site binding pocket of the enzyme, including Trp58, Trp59, Tyr62, His101, Leu165, Asp197, Glu233, and Asp300 [56,57,58].

Analysis of the docking data for natural xanthone **1** showed hydrogen bond interactions with carboxylate groups of the amino acid of the catalytic triad (Asp197, Glu233, and Asp300), suggesting competitive-type inhibition. Regarding compounds **10c** and **10f**, the dibenzo-γ-pyrone system of xanthone displays hydrophobic π-π and π-alkyl interactions with residues Trp59 and Tyr62. The fragment 2-(4-chlorophenyl)butoxy imidazole at C-2 of the xanthone core is involved in hydrophobic interactions, such as π-π and π-alkyl (Tyr62, Leu162, His 201, and Ala198), alkyl (Leu162, Ile235, Lys200), and π-anion (Asp300) interactions. In addition, unconventional hydrogen bonding interactions are observed with Asp197 and Asp300 residues of the catalytic triad. This could be due to the fact that compounds **10c** (Appendix A) and **10f** adopted U-shaped and V-shaped conformations, respectively, modifying their rearrangement in space and generating these interactions close to the catalytic pocket, thus suggesting competitive-type inhibition [55]. Data of **6c** and **9a** are summarized in Appendix A. 

### 2.7. Prediction of Drug-like Properties

The physicochemical properties of the synthesized compounds were analyzed using the OSIRIS DataWarrior program [59] and are summarized in Appendix A. Almost all compounds (except **10f** and **12f**) have a molecular weight under 500 g/mol. Lipophilicity is expressed as log *p* and represents the affinity of a molecule or a moiety for a lipophilic environment. Values close to 5 indicate high permeability of lipids, while negative values evidence low permeability [60]. 

Compounds **1**, **6a**, **7**, **9a**, **11b**, and **12b–e** showed acceptable log *p* values within the range of 2.78–4.96. Slight to moderate solubility in water was found for all compounds, with log S values ranging from −4.64 to −7.84 [61]. Acarbose (**14**) has a value of 0.58 (its polar groups form hydrogen bonds with water). The polar surface area (PSA) of a molecule is the sum of the surfaces of oxygen or nitrogen atoms and the hydrogen atoms attached to them. For a drug to cross the blood–brain barrier, the PSA value must be less than 90 A^2^ [62]. All derivatives herein evaluated met this requirement. Since **1**, **6a**, **6c**, **7**, **9a**, **10c**, **11b**, and **12b–e** comply with Lipinski’s rule of five (Appendix A), they are expected to have oral bioavailability.

### 2.8. Prediction of Druglikeness, ADME Properties, and Toxicity

The most potent α-glucosidase and α-amylase inhibitors (**6c**, **6e**, **9a**, **10c**, and **10f**) were analyzed with the online software PreADMET 2.0 to predict their druglikeness, ADME, and toxicity (Table 4) [63]. All compounds except **10f** complied with Lipinski’s rule of five. Regarding the intestinal barrier, represented by Caco-2 cells, the permeability of **6c**, **6e**, **10c**, and **10f** was moderate, while that of **9a** was poor. All five compounds have good human intestinal absorption (HIA) and acceptable permeability of the blood–brain barrier (BBB) and skin. Compounds **6e**, **10c**, and **10e** are non-mutagenic, and **6c**, **6e**, **10c**, and **10f** are non-carcinogenic on mice and rats. Compound **9a** had a carcinogenic effect on rats but not on mice. Furthermore, there was a medium risk of cardiotoxicity (hERG inhibition) for all five compounds when examined in silico. It is worth mentioning that the biological activity data, together with in silico studies (docking, drug prediction, and ADMET), allow for a broader perspective of the effects that the 4′-chlorophenylacetophenone and 4-bromobutoxy moieties produce on the glucosidase and amylase targets, suggesting structural features for the design of new molecules with antidiabetic properties.

## 3. Materials and Methods

### 3.1. General Information

Melting points were determined on electrothermal apparatus and are uncorrected. ^1^H-NMR spectra were recorded on Varian Mercury (300 MHz), Varian VNMR (500 MHz), Bruker 600AVANCE III (600 MHz), and Bruker Ascend (750 MHz) spectrometers. The chemical shifts (δ) are expressed in ppm relative to TMS as an internal standard. Multiplicities were denoted as follows: singlet (s), doublet (d), triplet (t), quartet (q), quintet (qu), multiplet (m), doublet of doublets (dd), double doublet of doublets (ddd), doublet of triplets (dt), double of quartets (dq), triplet of doublets (td), triplet of doublets multiplet (tdm), broad singlet (brs), broad doublet (brd), broad triplet (brt), and broad triplet of doublets (brtd). High-resolution mass spectra (HRMS, in electron ionization mode) were acquired on a Jeol JSM-GCMatell instrument. Electrospray mass spectra (ESI-MS) were captured on a micrOTOf-Q II spectrometer. Anhydrous solvents were obtained by a distillation process. Thin-layer chromatography was carried out on precoated silica gel plates (Merck 60F_254_). Silica gel (230–400 mesh) was used for flash chromatography. All standard reagents employed in synthesis and experiments were purchased from Sigma-Aldrich (acarbose (**14**), α-glucosidase from *Saccharomyces cerevisiae*, and α-amylase from porcine pancreas). 

### 3.2. Chemistry

2-Methoxybenzen-1,4-diol (**2**). A mixture of vanillin (1.0 mol equiv.) and MCPBA (77%, 1.4 mol equiv.) in CH_2_Cl_2_ (100 mL) was stirred at room temperature (rt) for 5 h. The reaction mixture was filtered and the solvent was removed under vacuum. The dried crude product was dissolved in MeOH (80 mL), and 6 N HCl (4 mL) was added. The mixture was then stirred at rt for 1 h, followed by removal of the solvent under vacuum. The residue was extracted with EtOAc (3 × 50 mL), the organic layers were dried (Na_2_SO_4_), and the solvent was evaporated under vacuum. The residue was purified by column chromatography over silica gel (*n*-hexane/EtOAc, 7:3), providing **2** as a brown solid (86%). *Rf* 0.20 (*n*-hexane/EtOAc, 7:3); mp 174–175 °C (Lit. 173–175 °C [28]). ^1^H-NMR (300 MHz, DMSO-*d*_6_) δ: 3.71 (s, 3H, OC*H*_3_), 6.17 (dd, *J* = 8.4, 2.7 Hz, 1H, H-5), 6.34 (d, *J* = 2.7 Hz, 1H, H-3), 6.59 (d, *J* = 8.7 Hz, 1H, H-6), 7.44 (s, 1H, OH), 8.37 (s, 1H, OH). ^13^C-NMR (75 MHz, DMSO-*d*_6_) δ: 55.3 (OCH_3_), 100.0 (C-3), 106.1 (C-5), 115.0 (C-6), 138.6 (C1), 147.6 (C2), 150.1 (C4). 

(2,5-dihydroxy-4-methoxyphenyl)(2-iodophenyl)methanone (**3a**). BF_3_
^•^ OEt_2_ (2.0 mol equiv.) was added to a solution of phenol **2** (2.0 mol equiv.) and 2-iodobenzoyl chloride (1.5 mol equiv.) under nitrogen atmosphere at 0 °C. The mixture was stirred at 80 °C for 3 h. The residue was poured into ice water (10 mL), adjusted to neutral pH with an aqueous saturated solution of NaHCO_3_, and extracted with EtOAc (3 × 30 mL). The organic layer was dried (Na_2_SO_4_) and concentrated under vacuum. The residue was purified by flash chromatography over silica gel (*n*-hexane/EtOAc, 9:1), providing **3a** as a yellow oil (81%). *Rf* 0.55 (*n*-hexane/EtOAc, 6:4). ^1^H-NMR (500 MHz, CDCl_3_) δ: 3.68 (s, 3H, OC*H*_3_), 6.42 (brs, 1H, OH-4), 6.52 (s, 1H, H-6), 6.60 (s, 1H, H-3), 7.18 (td, *J* = 8.0, 1.5 Hz, 1H, H-4′), 7.29 (dd, *J* = 8.0, 1.5 Hz, 1H, H-6′), 7.47 (td, *J* = 8.0, 1.5 Hz, 1H, H-5′), 7.93 (d, *J* = 8.0 Hz, 1H, H-3′), 12.26 (s, 1H, OH-C-2). ^13^C-NMR (125 MHz, CDCl_3_) δ: 56.2 (OCH_3_), 92.0 (C-2′), 99.7 (C-3), 110.9 (C-1), 113.1 (C-6), 127.90 (C-5′), 127.95 (C-6′), 131.0 (C-4′), 139.9 (C-3′ and C-5), 143.6 (C-1′), 154.7 (C-4), 161.8 (C-2), 200.0 (C=O). HRMS (EI^+^) calculated for C_14_H_11_IO_4_: 369.9702. Found: 369.9705. 

Methyl 2-(4-hydroxy-5-(2-iodobenzoyl)-2-methoxyphenoxy)acetate (**3b**). A solution of benzophenone **3a** (1.0 mol equiv.) and K_2_CO_3_ (1.5 mol equiv.) in dry acetone (7 mL) was stirred to 25 °C for 15 min, and methyl bromoacetate (**4a**) (1.5 mol equiv.) was added dropwise. The reaction mixture was refluxed at 60 °C for 3 h. The crude reaction mixture was filtered, and the solvent removed under vacuum. The residue was purified by flash chromatography over silica gel (*n*-hexane/EtOAc, 8:2), producing **3a** as a yellow oil (75%). *Rf* 0.45 (*n*-hexane/EtOAc, 6:4). ^1^H-NMR (500 MHz, CDCl_3_) δ: 3.70 (s, 3H, CO_2_OC*H*_3_), 3.94 (s, 3H, OC*H*_3_-2′), 4.45 (s, 2H, OC*H*_2_), 6.55 (s, 1H, H-3′), 6.62 (s, 1H, H-6′), 7.20 (td, *J* = 7.5, 1.5 Hz, 1H, H-4″), 7.26 (dd, *J* = 7.5, 1.5 Hz, 1H, H-6″), 7.47 (td, *J* = 7.5, 1.0 Hz, 1H, H-5″), 7.92 (d, *J* = 8.0 Hz, 1H, H-3″), 12.33 (s, 1H, OH). ^13^C-NMR (125 MHz, CDCl_3_) δ: 52.1 (CO_2_*C*H_3_), 56.3 (O*C*H_3_-2′), 67.5 (O*C*H_2_), 92.0 (C-2″), 101.0 (C-3′), 111.0 (C-5′), 119.1 (C-6′), 127.8 (C-5″), 128.0 (C-6″), 131.1 (C-4″), 139.5 (C-3″), 139.9 (C-1′), 142.0 (C-1″), 158.4 (C-2′), 162.4 (C-4′) 169.0 (*C*O_2_CH_3_), 200.2 (*C*O). HRMS (EI^+^) calculated for C_17_H_15_IO_6_: 441.9913. Found: 441.9917.

Methyl (Z)-3-(dimethylamino)-2-((3-methoxy-9-oxo-9H-xanthen-2-yl)oxy)acrylate (**5**). Methyl 2-((3-methoxy-9-oxo-9H-xanthen-2-yl)oxy)acetate (**6a**). A mixture of **3b** (1.0 mol equiv.) and DMADMF (4.0 mol equiv.) was placed at rt in a threaded ACE glass pressure tube with a sealed Teflon screw cap and heated at 120 °C for 48 h. The reaction mixture was cooled and diluted with CH_2_Cl_2_ (30 mL), and the solvent was then removed under vacuum. The residue was purified by flash chromatography over silica gel (n-hexane/EtOAc, 8:2), affording **6a** as a pale-yellow solid (20%) and **5** as a yellow solid (40%). **6a**: Rf 0.60 (n-hexane/EtOAc, 1:2); mp 139–140 °C. **5**: Rf 0.42 (n-hexane/EtOAc, 1:2); mp 164–165 °C. Data for **5**: ^1^H-NMR (500 MHz, CDCl_3_) δ: 2.98 (s, 6H, N(CH_3_)_2_), 3.61 (s, 3H, CO_2_CH_3_), 4.03 (s, 3H, OCH_3_-3′), 6.95 (s, 1H, H-4′), 7.22 (s, 1H, H-3), 7.35 (ddd, *J* = 8.0, 7.0, 1.0 Hz, 1H, H-7′), 7.44 (d, *J* = 8.5 Hz, 1H, H-5′), 7.67 (ddd, *J* = 8.5, 7.0, 1.5 Hz, 1H, H-6′), 7.71 (s, 1H, H-1′), 8.31 (dd, *J* = 8.0, 1.5 Hz, 1H, H-8′). ^13^C-NMR (125 MHz, CDCl_3_) δ: 42.2 (N(CH_3_)_2_), 51.0 (CO_2_CH_3_), 56.5 (OCH_3_-3′), 100.1 (C-4′), 109.0 (C-1′), 115.0 (C-9a’), 117.7 (C-5′), 121.5 (C-2, C-8a’), 123.5 (C-7′), 126.5 (C-8′), 133.9 (C-6′), 139.9 (C-3), 146.4 (C-2′), 152.8 (C-4a’), 155.2 (C-3′), 156.0 (C-4b’), 165.6 (CO_2_CH_3_), 176.0 (CO-9′). HMRS (EI^+^) calculated for C_20_H_19_NO_6_: 369.1212. Found: 369.1199.

Data for **6a**: ^1^H-NMR (500 MHz, CDCl_3_) δ: 3.83 (s, 3H, CO_2_C*H*_3_), 4.03 (s, 3H, OC*H*_3_-3′), 4.82 (s, 2H, OC*H*_2_, H-2), 6.95 (s, 1H, H-4′), 7.37 (td, *J* = 7.5, 1.0 Hz, 1H, H-7′), 7.45 (d, *J* = 8.5 Hz, 1H, H-5′), 7.60 (s, 1H, H-1′), 7.68 (ddd, *J* = 8.5, 7.5, 1.7 Hz, 1H, H-6′), 8.31 (dd, *J* = 7.5, 1.7 Hz, 1H, H-8′). ^13^C-NMR (125 MHz, CDCl_3_) δ: 52.3 (CO_2_*C*H_3_), 56.5 (O*C*H_3_-3′), 65.9 (O*C*H_2_), 100.2 (C-4′), 107.4 (C-1′), 114.7 (C-9a’), 117.7 (C-5′), 121.5 (C-8a’), 123.8 (C-7′), 126.5 (C-8′), 134.1 (C-6′), 144.9 (C-2′), 153.0 (C-4a’), 155.8 (C-3′), 156.1 (C-4b’), 168.6 (*C*O_2_CH_3_), 175.9 (*C*O-9′). HRMS (EI^+^) calculated for C_17_H_14_O_6_: 314.0790. Found: 314.0790.

2-Hydroxy-3-methoxy-9H-xanthen-9-one (**1**). A mixture of benzophenone **3a** (1.0 mol equiv.), and KOH (3.0 mol equiv.) in distilled water (7 mL) was placed at rt in a threaded ACE glass pressure tube with a sealed Teflon screw cap and was heated at 100 °C for 6 h. An aqueous solution of HCl (10%) was added until neutral and the residue was extracted with EtOAc (3 × 50 mL). The organic layer was dried (Na_2_SO_4_) and concentrated under vacuum. The crude product was purified through recrystallization with EtOAc, furnishing **1** as colorless crystals (93%). *Rf* 0.5 (*n*-hexane/EtOAc, 1:1); mp 170–171 °C (lit. 173–175 °C [28]). ^1^H-NMR (300 MHz, DMSO-*d_6_*) δ: 3.92 (s, 3H, OC*H*_3_), 6.86 (s, 1H, H-4), 7.28 (td, *J =* 8.1, 0.9 Hz, 1H, H-7), 7.39 (d, *J* = 8.4 Hz, 1H, H-5), 7.52 (s, 1H, H-1), 7.61 (td, *J* = 8.4, 1.5 Hz, 1H, H-6), 8.18 (dd, *J* = 8.1, 1.5 Hz, 1H, H-8), 9.30 (s, 1H, OH). ^13^C-NMR (75 MHz, CDCl_3_) δ: 55.8 (O*C*H_3_), 99.1 (C-4), 108.4 (C-1), 114.4 (C-9a), 117.2 (C-5), 120.7 (C-8a), 123.0 (C-7), 125.5 (C-8), 133.4 (C-6), 143.8 (C-2), 150.8 (C-4a), 154.3 (C-3), 155.4 (C-4b), 175.1 (*C*O-9). HRMS (ESI^+^) calculated for C_14_H_10_O_4_ + Na: 265.0477 ([M+Na]^+^). Found: 265.0472 [M+Na]^+^.

General method for preparing 2-alkoxy-substituted xanthones **6a–d**

A solution of 2-hydroxyxanthone **1** (1.0 mol equiv.) and K_2_CO_3_ (1.5 mol equiv.) in dry acetone (7 mL) was stirred at 25 °C for 15 min, and α-halocarbonyl (**4a–c**) or allyl bromide (**4d**) (1.5 mol equiv.) was added dropwise. The reaction mixture was refluxed at 60 °C for 3 h. After the reaction was completed (as monitored by TLC), the reaction mixture was filtered, and the solvent was removed under vacuum. The residue was purified by flash chromatography over silica gel (*n*-hexane/EtOAc, 8:2), generating the corresponding product.

Methyl 2-((3-methoxy-9-oxo-9H-xanthen-2-yl)oxy)acetate (**6a**). A pale-yellow solid (87%), *Rf* 0.60 (*n*-hexane/EtOAc, 1:2); mp 139–140 °C. 

3-Methoxy-2-(2-oxopropoxy)-9H-xanthen-9-one (**6b**). A white solid (92%), *Rf* 0.50 (*n*-hexane/EtOAc, 7:3); mp 144–145 °C. ^1^H-NMR (500 MHz, CDCl_3_) δ: 2.33 (s, 3H, H-3′), 4.03 (s, 3H, C*H*_3_CO), 4.74 (s, 2H, C*H*_2_), 6.95 (s, 1H, H-4), 7.37 (td, *J* = 7.2, 0.8 Hz, 1H, H-7), 7.45 (d, *J* = 8.4 Hz, 1H, H-5), 7.56 (brs, 1H, H-1), 7.69 (td, *J* = 8.4, 1.6 Hz, 1H, H-6), 8.32 (dd, *J* = 8.0, 1.6 Hz, 1H, H-8′). ^13^C-NMR (125 MHz, CDCl_3_) δ: 26.5 (*C*H_3_CO), 56.5 (O*C*H_3_), 73.7 (*C*H_2_), 100.2 (C-4), 107.5 (C-1), 114.7 (C-9a), 117.6 (C-5), 121.4 (C-8a), 123.8 (C-7), 126.5 (C-8), 134.1 (C-6), 144.9 (C-2), 153.0 (C-4a), 155.8 (C-3), 156.6 (C-4b), 175.8 (*C*O-9), 203.7 (CH_3_*C*O). HRMS (ESI^+^) calculated for C_17_H_14_O_5_ + H: 299.0919 ([M+H]^+^). Found: 299.0893 [M+H]^+^.

2-(2-(4-Chlorophenyl)-2-oxoethoxy)-3-methoxy-9H-xanthen-9-one (**6c**). A white solid (98%), *Rf* 0.51 (*n*-hexane/EtOAc, 7:3); mp 175–176 °C. ^1^H-NMR (750 MHz, DMSO-*d_6_*) δ: 3.99 (s, 3H, OC*H*_3_), 5.74 (s, 2H, C*H*_2_), 7.26 (s, 1H, H-4), 7.44 (td, *J* = 7.5, 0.7 Hz, 1H, H-7), 7.48 (s, 1H, H-1), 7.62 (d, *J* = 8.2 Hz, 1H, H-5), 7.66–7.68 (m, 2H, H-3″), 7.82 (td, *J* = 8.2, 1.5 1H, H-6), 8.07–8.09 (m, 2H, H-2″), 8.14 (dd, *J* = 8.2, 1.5 Hz, 1H, H-8). RMN-^13^C (187.5 MHz, CDCl_3_) δ: 56.5 (O*C*H_3_), 76.8 (*C*H_2_), 100.6 (C-4), 106.6 (C-1), 113.7 (C-9a), 117.9 (C-5), 120.7 (C-8a), 124.1 (C-7), 125.7 (C-8), 129.0 (C-3″), 129.9 (C-2″), 132.9 (C-4″), 134.7 (C-6), 138.8 (C-1″), 145.0 (C-2), 152.1 (C-4a), 155.4 (C-3), 155.6 (C-4b), 174.5 (*C*O-9), 193.3 (*C*O-2′). HRMS (EI^+^) calculated for C_22_H_15_ClO_5_: 394.0608. Found: 394.0610.

2-(Allyloxy)-3-methoxy-9H-xanthen-9-one (**6d**). A white solid (87%), *Rf* 0.45 (*n*-hexane/EtOAc, 7:3); mp 150–151 °C. ^1^H-NMR (750 MHz, DMSO-*d_6_*) δ: 3.94 (s, 3H, OC*H*_3_), 4.65 (d, *J* = 5.2 Hz, 2H, H-1′), 5.29 (dd, *J* = 10.5, 1.5 Hz, 1H, H-3′), 5.44 (dd, *J* = 17.2, 1.5 Hz, 1H, H-3′), 6.08 (m, 1H, H-2′), 7.16 (s, 1H, H-4), 7.460 (brt, *J* = 7.5 Hz, 1H, H-7), 7.463 (s, 1H, H-1), 7.56 (d, *J* = 8.2 Hz, 1H, H-5), 7.79 (td, *J* = 8.2, 0.7 Hz, 1H, H-6), 8.14 (d, *J* = 7.5 Hz, 1H, H-8). ^13^C-NMR (187.5 MHz, DMSO-*d_6_*) δ: 56.4 (O*C*H_3_), 69.1 (C-1′), 100.3 (C-1), 106.1 (C-4), 113.8 (C-9a), 117.80 (C-5), 117.87 (C-3′), 120.7 (C-8a), 124.0 (C-7), 125.7 (C-8), 133.3 (C-2′), 134.5 (C-6), 145.3 (C-2), 151.8 (C-4a), 155.4 (C-4b), 155.6 (C-3), 174.5 (*C*O-9). HRMS (ESI^+^) calculated for C_17_H_14_O_4_ + H: 283.0970 ([M+H]^+^). Found: 283.0944 [M+H]^+^.

1-Allyl-2-hydroxy-3-methoxy-9H-xanthen-9-one (**7**). A solution of **6d** (1.0 mol equiv.) in decalin (1.0 mL) was placed at rt in a threaded ACE glass pressure tube with a sealed Teflon screw cap and heated at 200 °C for 12 h. The reaction was cooled and diluted with CH_2_Cl_2_ (30 mL), and the solvent was removed under vacuum. The residue was purified by flash chromatography over silica gel (*n*-hexane/EtOAc, 8:2), resulting in **7** as a yellow solid (95%). *Rf* 0.60 (*n*-hexane/EtOAc, 6:4); mp 167–169 °C. ^1^H-NMR (750 MHz, CDCl_3_) δ: 4.03 (s, 3H, OC*H*_3_), 4.26 (dt, *J* = 6.0, 1.6 Hz, 2H, H-1′), 5.00 (dq, *J* = 10.0, 1.6 Hz, 1H, H-3′), 5.08 (dq, *J* = 17.2, 1.6 Hz, 1H, H-3′), 5.67 (s, 1H, OH), 6.18 (ddt, *J* = 17.2, 10.0, 6.0 Hz, 1H, H-2′), 6.85 (s, 1H, H-4), 7.32 (td, *J* = 7.2, 0.8 Hz, 1H, H-7), 7.38 (dd, *J* = 8.4, 0.8 Hz, 1H, H-5), 7.64 (td, *J* = 7.2, 1.6 Hz, 1H, H-6), 8.29 (dd, *J* = 8.0, 2.0 Hz, 1H, H-8). ^13^C-NMR (187.5 MHz, CDCl_3_) δ: 30.2 (C-1′), 56.3 (O*C*H_3_), 97.6 (C-4), 113.9 (C-9a), 114.6 (C-3′), 117.0 (C-5), 122.5 (C-8a), 123.5 (C-7), 125.7 (C-1), 126.7 (C-8), 133.6 (C-6), 136.8 (C-2′), 140.6 (C-2), 151.5 (C-3), 153.0 (C-4a), 155.1 (C-4b), 177.6 (*C*O-9). HRMS (ESI^+^) calculated for C_17_H_14_O_4_ + H: 283.0970 ([M+H]^+^). Found: 283.0958 [M+H]^+^.

1-Allyl-2-(2-(4-chlorophenyl)-2-oxoethoxy)-3-methoxy-9H-xanthen-9-one (**6e**). Following the method for preparing **6c** with **7** as a starting material, **6e** was obtained as a white solid (97%). *Rf* 0.67 (*n*-hexane/EtOAc, 7:3); mp 197–198 °C. ^1^H-NMR (600 MHz, CDCl_3_) δ: 3.93 (s, 3H, OC*H*_3_), 4.30 (dt, *J* = 5.7, 1.2 Hz, 2H, H-1′), 4.94 (dq, *J* = 16.8, 1.8 Hz, 1H, H-3′), 4.96 (dq, *J* = 9.0, 1.8 Hz, 1H, H-3′), 5.15 (s, 2H, H-1″), 6.16 (ddt, *J* = 16.8, 9.0, 5.7 Hz, 1H, H-2′), 6.87 (s, 1H, H-4), 7.34 (brt, *J* = 7.8 Hz, 1H, H-7), 7.40 (brd, *J* = 7.8, 0.6 Hz, 1H, H-5), 7.46–7.48 (m, 2H, H-3‴), 7.66 (td, *J* = 8.4, 1.2 1H, H-6), 7.96–7.98 (m, 2H, H-2″); 8.27 (dd, *J* = 7.8, 1.8 Hz, 1H, H-8). RMN-^13^C (150 MHz, CDCl_3_) δ: 31.0 (C-1′), 56.0 (O*C*H_3_), 75.7 (C-1″), 99.0 (C-4), 113.6 (C-9a), 114.6 (C-3′), 117.0 (C-5), 122.4 (C-8a), 123.7 (C-7), 126.7 (C-8), 129.0 (C-3″), 129.5 (C-2″), 133.1 (C-1‴), 133.94 (C-6), 135.6 (C-1), 137.7 (C-2′), 140.0 (C-4‴), 142.7 (C-2), 155.0 (C-4b), 155.8 (C-4a), 157.3 (C3), 177.2 (*C*O-9), 193.3 (*C*O-2″). HRMS (EI^+^) calculated for C_25_H_19_ClO_5_: 434.0921. Found: 434.0926.

5-Methoxy-3,3-dimethylpyrano[3,2-a]xanthen-12(3H)-one (**8**). 1,1-Diethoxy-3-methylbut-1-ene (1.2 mol equiv.) was added dropwise to a solution of **1** (1.0 mol equiv.) and 3-methylpicoline (0.6 mol equiv.) in xylene (5 mL). The mixture was refluxed at 120 °C for 30 h and then filtered, and the solvent was removed under vacuum. The residue was purified by flash chromatography over silica gel (*n*-hexane/EtOAc, 9:1), providing **8** as a yellow solid (56%). *Rf* 0.50 (*n*-hexane/EtOAc, 7:3); mp 182–184 °C. ^1^H-NMR (750 MHz, DMSO-*d_6_*) δ: 1.40 (s, 6H, (C*H*_3_)_2_), 3.94 (s, 3H, OC*H*_3_), 5.93 (d, *J* = 10.1 Hz, 1H, H-2), 7.12 (s, 1H, H-6), 7.42 (brt, *J* = 8.2 Hz, 1H, H-10), 7.56 (d, *J* = 8.2 Hz, 1H, H-8), 7.79 (td, *J* = 8.2, 1.5 Hz, 1H, H-9), 7.96 (d, *J* = 10.1 Hz, 1H, H-1), 8.13 (dd, *J* = 8.2, 1.5 Hz, 1H, H-11). ^13^C-NMR (187.5 MHz, DMSO-*d_6_*) δ: 26.8 ((*C*H_3_)_2_), 56.2 (O*C*H_3_), 75.2 (C-3), 100.2 (C-6), 109.0 (C-12a), 117.3 (C-8), 119.2 (C-12b), 120.2 (C-1), 121.7 (C-11a), 123.9 (C-10), 125.8 (C-11), 132.4 (C-2), 134.5 (C-9), 138.7 (C-4a), 152.5 (C-6a), 154.2 (C-5), 154.6 (C-7a), 176.7 (*C*O). HRMS (EI^+^) calculated for C_19_H_16_O_4_: 308.1049. Found: 308.1045.

2-(4-Bromobutoxy)-3-methoxy-9H-xanthen-9-one (**9a**). A mixture of **1** (1.0 mol equiv.) and K_2_CO_3_ (2.5 mol equiv.) in dry acetone (10 mL) was stirred at 25 °C for 15 min, and 1,4-dibromobutane (7.5 mol equiv.) was then added dropwise. The reaction mixture was refluxed at 60 °C for 6 h. The residue was purified by flash chromatography over silica gel (*n*-hexane/EtOAc, 9:1) to provide **9a** as a white solid (95%). *Rf* 0.43 (*n*-hexane/EtOAc, 7:3); mp 116–117 °C. ^1^H-NMR (750 MHz, CDCl_3_) δ: 2.03–2.08 (m, 2H, H-2′), 2.09–2.14 (m, 2H, H-3′), 3.52 (t, *J* = 6.0 Hz, H-4′), 4.00 (s, 3H, OC*H*_3_), 4.17 (t, *J* = 6.0 Hz, H-1′), 6.91 (s, 1H, H-4), 7.37 (td, *J* = 7.5, 0.75 Hz, 1H, H-7), 7.46 (dd, *J* = 9.0, 0.75 Hz, 1H, H-5), 7.65 (s, 1H, H-1), 7.68 (td, *J* = 8.25, 1.5 Hz, 1H, H-6), 8.33 (dd, *J* = 7.5, 1.5 Hz, 1H, H-8). ^13^C-NMR (187.5 MHz, CDCl_3_) δ: 27.6 (C-2′), 29.4 (C-3′), 33.3 (C-4′), 56.4 (O*C*H_3_), 68.2 (C-1′), 99.7 (C-4), 106.6 (C-1), 114.8 (C-9a), 117.6 (C-5), 121.4 (C-8a), 123.7 (C-7), 126.5 (C-8), 133.9 (C-6), 145.9 (C-2), 152.4 (C-4a), 155.7 (C-3), 156.0 (C-4b), 176.0 (*C*O-9). HRMS (EI^+^) calculated for C_18_H_17_BrO_4_: 376.0310. Found: 376.0307.

1-Allyl-2-(4-bromobutoxy)-3-methoxy-9H-xanthen-9-one (**9b**). Following the method for preparing **9a** with **7** as a starting material, **9b** was obtained as a yellow solid (98%). *Rf* 0.56 (*n*-hexane/EtOAc, 7:3); mp 69–70 °C. ^1^H-NMR (600 MHz, CDCl_3_) δ: 1.97 (q, *J =* 6.6 Hz, 2H, H-2″), 2.15 (q, *J* = 6.6 Hz, 2H, H-3″), 3.46 (t, *J* = 6.6 Hz, 2H, H-4″), 3.95 (t, *J* = 6.0 Hz, 2H, H-1″), 3.98 (s, 3H, OC*H*_3_), 4.22 (dt, *J* = 6.0, 1.8 Hz, 2H, H-1′), 4.96 (dq, *J* = 12.0, 1.8 Hz, 1H, H-3′), 4.98 (dq, *J* = 17.0, 1.8 Hz, 1H, H-3′), 6.16 (ddt, *J* = 17.0, 12.0, 6.0 Hz, 1H, H-2′), 6.85 (s, 1H, H-4), 7.35 (td, *J* = 8.4, 0.6 Hz, 1H, H-7), 7.38 (dd, *J* = 8.4, 0.6 Hz, 1H, H-5), 7.64 (td, *J* = 8.4, 1.8 Hz, 2H, H-6), 8.27 (dd, *J* = 8.4, 1.8 Hz, 1H, H-8). ^13^C-NMR (150 MHz, CDCl_3_) δ: 28.8 (C-2″), 29.4 (C-3″), 30.9 (C-1′), 33.632 (C-4″), 56.0 (O*C*H_3_), 72.3 (C-1″), 98.8 (C-4), 113.5 (C-9a), 114.4 (C-3′), 116.9 (C-5), 122.5 (C-8a), 123.6 (C-7), 126.7 (C-8), 133.7 (C-6), 135.4 (C-1), 137.7 (C-2′), 143.3 (C-2), 155.0 (C-4a), 155.5 (C-4b), 158.0 (C-3), 177.1 (*C*O-9). HRMS (EI^+^) calculated for C_21_H_21_BrO_4_: 416.0623 [M^+^]. Found: 416.0589 [M^+^].

General method for preparing imidazole-substituted xanthones **10a–f**

Alkoxy-substituted xanthones **9a–b** (1.0 mol equiv.) were each added dropwise to a mixture of the respective imidazole **13a–c** (2.0 mol equiv.) and K_2_CO_3_ (10.0 mol equiv.) in dry acetone (10 mL) at rt, followed by stirring for 15 min. After subsequent refluxing at 60 °C for 48 h, the reaction mixture was filtered and the solvent was removed under vacuum. The residue was diluted with 50 mL of EtOAc, washed three times with brine, and dried over anhydrous Na_2_SO_4_. The solvent was removal under vacuum, and the residual mixture was purified by flash chromatography over silica gel (CH_2_Cl_2_/MeOH, 98:02), generating the corresponding product.

2-(4-(1H-imidazol-1-yl)butoxy)-3-methoxy-9H-xanthen-9-one (**10a**). A white solid (75%), *Rf* 0.73 (*n*-hexane/EtOAc, 1:2); mp 137–138 °C. ^1^H-NMR (750 MHz, DMSO-*d_6_*) δ: 1.69 (qu, *J* = 6.7 Hz, 2H, C*H*_2_, H-2′), 1.88 (qu, *J* = 6.7 Hz, C*H*_2_, H-3′), 3.93 (s, 3H, OC*H*_3_), 4.03–4.08 (m, 4H, H-1′, H-4′), 6.89 (s, 1H, H-4″), 7.18 (s, 1H, H-4), 7.21 (s, 1H, H-5″), 7.43 (td, *J* = 7.5, 0.7 Hz, 1H, H-7), 7.46 (s, 1H, H-1), 7.59 (brd, *J* = 8.2, 0.6 Hz, 1H, H-5), 7.67 (brs, 1H, H-2″), 7.81 (td, *J* = 8.2, 1.5 Hz, 1H, H-6), 8.15 (dd, *J* = 7.5, 1.5 Hz, 1H, H-8). ^13^C-NMR (187.5 MHz, DMSO-*d_6_*) δ: 25.5 (C-2′), 27.4 (C-3′), 45.6 (C-4′), 56.5 (O*C*H_3_), 68.1 (C-1′), 100.4 (C-1), 105.6 (C-4), 113.8 (C-9a), 117.8 (C-5), 119.3 (C-5″), 120.7 (C-8a), 124.1 (C-7), 125.7 (C-8), 128.3 (C-4″), 134.6 (C-6), 137.3 (C-2″), 146.8 (C-2), 151.8 (C-4a), 155.4 (C-4b), 155.6 (C-3), 174.6 (*C*O-9). HRMS (EI^+^) calculated for C_21_H_20_N_2_O_4_: 364.1423. Found: 364.1433.

3-Methoxy-2-(4-(2-phenyl-1H-imidazol-1-yl)butoxy)-9H-xanthen-9-one (**10b**). A white solid (40%), *Rf* 0.59 (CH_2_Cl_2_/MeOH, 9:1); mp 132–134 °C. ^1^H-NMR (500 MHz, DMSO-*d_6_*) δ: 1.67 (qu, *J* = 6.5 Hz, 2H, H-2′), 1.84 (qu, *J* = 7.0 Hz, 2H, H-3′), 3.90 (s, 3H, OC*H*_3_), 3.96 (t, *J* = 7.0 Hz, 2H, H-1′), 4.15 (t, *J* = 7.5 Hz, 2H, H-4′), 7.01 (brs, 1H, H-4″), 7.17 (s, 1H, H-4), 7.36 (brd, *J* = 0.5 Hz, 1H, H-5″), 7.39–7.46 (m, 4H, H-1, H-7, H-3‴), 7.57–7.62 (m, 3H, H-5, H-2‴), 7.65–7.72 (m, 1H, H-4‴), 7.81 (td, *J* = 8.5, 2.0 Hz, 2H, H-6), 8.16 (dd, *J* = 8.0, 1.5 Hz, 1H, H-8). ^13^C-NMR (125 MHz, DMSO-*d_6_*) δ: 25.4 (C-2′), 27.1 (C-3′), 45.8 (C-4′), 56.4 (O*C*H_3_), 67.8 (C-1′), 100.3 (C-4), 105.6 (C-1), 113.8 (C-9a), 117.8 (C-5), 120.7 (C-5″), 121.6 (C-8a), 124.0 (C-7), 125.7 (C-8), 127.8 (C-4″), 128.31 (C-2‴), 128.35 (C-3‴), 128.4 (C-1‴), 131.0 (C-4‴), 134.5 (C-6), 145.7 (C-2), 146.3 (C-2″), 151.8 (C-4a), 155.4 (C-4b), 155.6 (C-3), 174.5 (*C*O-9). HRMS (EI^+^) calculated for C_27_H_24_N_2_O_4_: 440.1736. Found: 440.1753.

2-(4-(2-(4-Chlorophenyl)-1H-imidazol-1-yl)butoxy)-3-methoxy-9H-xanthen-9-one (**10c**). A white solid (50%), *Rf* 0.62 (CH_2_Cl_2_/MeOH, 9:1); mp 155–157 °C. ^1^H-NMR (750 MHz, DMSO-*d_6_*) δ: 1.69 (qu, *J* = 6.7 Hz, 2H, H-2′), 1.86 (qu, *J* = 7.5 Hz, 2H, H-3′), 3.91 (s, 3 H, OC*H*_3_), 4.01 (t, *J* = 6.7 Hz, 2H, H-1′), 4.17 (t, *J* = 7.5 Hz, 2H, H-4′), 7.02 (d, *J* = 0.7 Hz, 1H, H-4″), 7.21 (s, 1H, H-4), 7.39 (dd, *J* = 0.7 Hz, 1H, H-5″), 7.42–7.50 (m, 4H, H-1, H-7, H-3‴), 7.60–7.65 (m, 3H, H-5, H-2‴), 7.83 (td, *J* = 8.2, 1.5 Hz, 1H, H-6), 8.18 (dd, *J* = 8.2, 1.5 Hz, 1H, H-8). ^13^C-NMR (187.5 MHz, DMSO-*d_6_*) δ: 25.4 (C-2′), 27.1 (C-3′), 45.9 (C-4′), 56.4 (O*C*H_3_), 67.8 (C-1′), 100.3 (C-4), 105.6 (C-1), 113.8 (C-9a), 117.8 (C-5), 120.8 (C-8a), 122.1 (C-5″), 124.1 (C-7), 125.7 (C-8), 128.0 (C-4″), 128.5 (C-3‴), 129.8 (C-1‴), 129.9 (C-2‴), 133.0 (C-4‴), 134.6 (C-6), 145.1 (C-2″), 145.7 (C-2), 151.8 (C-4a), 155.4 (C-4b), 155.6 (C-3), 174.6 (*C*O-9). HRMS (EI^+^) calculated for C_27_H_23_N_2_O_4_Cl: 474.1346. Found: 474.1375.

2-(4-(1H-Imidazol-1-yl)butoxy)-1-allyl-3-methoxy-9H-xanthen-9-one (**10d**). A white solid (78%), *Rf* 0.56 (CH_2_Cl_2_/MeOH, 9:1); mp 155–157 °C. ^1^H-NMR (600 MHz, DMSO-*d_6_*) δ: 1.66 (qu, *J* = 7.2 Hz, 2H, H-2″), 1.87–1.93 (m, 2H, H-3″), 3.85 (t, *J =* 6.6 Hz, 2H, H-1″), 3.95 (s, 3H, OC*H*_3_), 4.05 (t, *J* = 7.2 Hz, 2H, H-4″), 4.09 (brd, *J* = 6.0 Hz, 2H, H-1′), 4.86–4.91 (m, 2H, H-3′), 5.99 (ddt, *J* = 16.8, 10.8, 6.0 Hz, 1H, H-2′), 6.92 (brs, 1H, H-4‴), 7.14 (s, 1H, H-4), 7.21 (brs, 1H, H-5‴), 7.41 (td, *J* = 7.8, 0.6 Hz, 1H, H-7), 7.54 (brd, *J* = 8.4 Hz, 1H, H-5), 7.67 (brs, 1H, H-2‴), 7.78 (td, *J* = 8.4, 1.2 Hz, 1H, H-6), 8.11 (dd, *J* = 7.8, 1.2 Hz, 1H, H-8). ^13^C-NMR (150 MHz, DMSO-*d_6_*) δ: 26.7 (C-2″), 27.3 (C-3″), 30.3 (C-1′), 45.7 (C-4″), 56.4 (O*C*H_3_), 72.2 (C-1″), 99.6 (C-4), 112.3 (C-9a), 114.5 (C-3′), 117.2 (C-5), 119.3 (C-5‴), 121.7 (C-8a), 124.0 (C-7), 126.7 (C-8), 128.4 (C-4‴), 134.0 (C-1), 134.5 (C-6), 137.5 (C-2′, C-2‴), 142.9 (C-2), 154.4 (C-4b), 155.0 (C-4a), 158.0 (C-3), 176.0 (*C*O-9). HRMS (EI^+^) calculated for C_24_H_24_N_2_O_4_: 404.1736. Found: 404.1744.

1-Allyl-3-methoxy-2-(4-(2-phenyl-1H-imidazol-1-yl)butoxy)-9H-xanthen-9-one (**10e**). A yellow oil (12%), *Rf* 0.63 (CH_2_Cl_2_/MeOH, 9:1). ^1^H-NMR (500 MHz, CDCl_3_) δ: 1.71–1.80 (m, 2H, H-2″), 1.97–2.08 (m, 2H, H-3″), 3.87 (t, *J =* 6.0 Hz, 2H, H-1″), 3.92 (s, 3H, OC*H*_3_), 4.10–4.22 (m, 4H, H-1′, H-4″), 4.90 (dq, *J* = 10.0, 1.5 Hz, 1H, H-3′), 4.95 (dq, *J* = 17.0, 1.5 Hz, 1H, H-3′), 6.12 (ddt, *J* = 17.0, 10.0, 6.0 Hz, 1H, H-2′), 6.84 (s, 1H, H-4), 7.09 (s, 1H, H-4‴), 7.17 (s, 1H, H-5‴), 7.33 (td, *J* = 8.0, 1.0 Hz, 1H, H-7), 7.36–7.47 (m, 4H, H-5, H-3^IV^, H-4^IV^), 7.60 (dd, *J* = 7.0, 1.0 Hz, 2H, H-2^IV^), 7.65 (td, *J* = 8.5, 1.0 Hz, 1H, H-6), 8.27 (dd, *J* = 8.0, 2.0 Hz, 1H, H-8). ^13^C-NMR (125 MHz, CDCl_3_) δ: 27.1 (C-2″), 27.8 (C-3″), 30.8 (C-1′), 46.6 (C-4″), 55.9 (O*C*H_3_), 72.5 (C-1″), 98.8 (C-4), 113.5 (C-9a), 114.4 (C-3′), 117.0 (C-5), 120.3 (C-5‴), 122.4 (C-8a), 123.6 (C-7), 125.2 (C-8), 126.7 (C-4‴), 128.6 (C-2^IV^), 128.7 (C-4^IV^), 128.9 (C-3^IV^), 130.9 (C-1^IV^), 133.8 (C-1), 135.3 (C-6), 137.7 (C-2′), 143.2 (C-2), 147.7 (C-2‴), 155.0 (C-4b), 155.5 (C-4a), 157.8 (C-3), 177.1 (*C*O-9). HRMS (EI^+^) calculated for C_30_H_28_O_4_N_2_: 480.2049. Found: 480.2050.

1-Allyl-2-(4-(2-(4-chlorophenyl)-1H-imidazol-1-yl)butoxy)-3-methoxy-9H-xanthen-9-one (**10f**). A brown solid (37%), *Rf* 0.70 (CH_2_Cl_2_/MeOH, 9:1); mp 90–92 °C. ^1^H-NMR (750 MHz, DMSO-*d_6_*) δ: 1.59 (qu, *J* = 6.7 Hz, 2H, H-2″), 1.84 (qu, *J* = 6.7 Hz, 2H, H-3″), 3.76 (t, *J =* 6.7 Hz, 2H, H-1″), 3.90 (s, 3H, OC*H*_3_), 4.03 (brd, *J* = 6.0 Hz, 2H, H-1′), 4.16 (t, *J* = 6.7 Hz, 2H, H-4″), 4.82 (dq, *J* = 18.0, 1.5 Hz, 1H, H-3′), 4.85 (dq, *J* = 10.5, 1.5 Hz, 1H, H-3′), 5.93 (ddt, *J* = 18.0, 10.5, 6.0 Hz, 1H, H-2′), 7.03 (d, *J* = 0.7 Hz, 1H, H-4‴), 7.12 (s, 1H, H-4), 7.37 (d, *J* = 0.7 Hz, 1H, H-5‴), 7.41 (td, *J* = 7.5, 0.7 Hz, 1H, H-7), 7.48–7.52 (m, 2H, H-3^IV^), 7.54 (d, *J* = 8.2 Hz, 1H, H-5), 7.62–7.66 (m, 2H, H-2^IV^), 7.78 (td, *J* = 8.2, 1.5 Hz, 1H, H-6), 8.11 (dd, *J* = 8.2, 1.5 Hz, 1H, H-8). ^13^C-NMR (187.5 MHz, DMSO-*d_6_*) δ: 26.5 (C-2″), 27.0 (C-3″), 30.3 (C-1′), 46.1 (C-4″), 56.3 (O*C*H_3_), 72.0 (C-1″), 99.6 (C-4), 112.3 (C-9a), 114.4 (C-3′), 117.2 (C-5), 121.7 (C-8a), 122.0 (C-5‴), 124.0 (C-7), 126.0 (C-8), 128.0 (C-4‴), 128.5 (C-3^IV^), 129.9 (C-1^IV^), 130.0 (C-2^IV^), 133.1 (C-4^IV^), 133.9 (C-1), 134.5 (C-6), 137.6 (C-2′), 142.8 (C-2), 145.3 (C-2‴), 154.4 (C-4b), 154.9 (C-4a), 157.9 (C-3), 176.0 (*C*O-9). HRMS (EI^+^) calculated for C_30_H_27_N_2_O_4_Cl: 514.1659. Found: 514.1658.

General procedure for preparing 2-alkoxy-substituted xanthones **11a–b**

Epichlorohydrin (2.5 mol equiv.) was added dropwise to a solution of xanthone **1** or **7** (1.0 mol equiv.) and KOH (2.5 mol equiv.) in EtOH (7 mL), followed by stirring for 15 min. After the subsequent refluxing of the reaction mixture for 12 h, the residual mixture was filtered, and the solvent was removed under vacuum. The crude was purified by flash chromatography over silica gel (*n*-hexane/EtOAc, 8:2).

3-Methoxy-2-(oxiran-2-ylmethoxy)-9H-xanthen-9-one (**11a**). A white solid (96%), *Rf* 0.16 (*n*-hexane/EtOAc, 7:3); mp 143–144 °C. ^1^H-NMR (500 MHz, CDCl_3_) δ: 2.81 (dd, *J* = 4.5, 2.5 Hz, 1H, H-3′), 2.94 (m, 1H, H-3′), 3.44–3.48 (m, 1H, H-2′), 4.00 (s, 3H, OC*H*_3_), 4.01 (dd, *J* = 11.0, 6.0 Hz, 1H, H-1′), 4.40 (dd, *J* = 11.0, 3.0 Hz, 1H, H-1′), 6.91 (s, 1H, H-4), 7.36 (td, *J* = 7.5, 0.75 Hz, 1H, H-7), 7.44 (dd, *J* = 8.5, 0.75 Hz, 1H, H-5), 7.65–7.70 (m, 2H, H-1, H-6), 8.32 (dd, *J* = 8.0, 2.0 Hz, 1H, H-8). ^13^C-NMR (125 MHz, CDCl_3_) δ: 44.7 (C-3′), 49.7 (C-2′), 56.3 (O*C*H_3_), 70.1 (C-1′), 99.8 (C-4), 107.1 (C-1), 114.7 (C-9a), 117.6 (C-5), 121.4 (C-8a), 123.7 (C-7), 126.4 (C-8), 133.9 (C-6), 145.6 (C-2), 152.6 (C-4a), 155.7 (C-3), 156.0 (C-4b), 175.9 (*C*O-9). HRMS (EI^+^) calculated for C_17_H_14_O_5_: 298.0841. Found: 298.0842.

1-Allyl-3-methoxy-2-(oxiran-2-ylmethoxy)-9H-xanthen-9-one (**11b**). A white solid (98%), *Rf* 0.34 (*n*-hexane/EtOAc, 7:3); mp 115–116 °C. ^1^H-NMR (500 MHz, CDCl_3_) δ: 2.73 (dd, *J =* 5.0, 2.5 Hz, 1H, H-3″), 2.89 (dd, *J* = 5.0, 4.5 Hz, 1H, H-3″), 3.39–3.42 (m, 1H, H-2″), 3.96–4.00 (m, 4H, H-1″, OC*H*_3_), 4.15 (dd, *J* = 11.0, 3.5 Hz, 1H, H-1″), 4.23–4.31 (m, 2H, H-1′), 4.98 (dq, *J* = 11.5, 2.0 Hz, 1H, H-3′), 4.99–5.02 (m, 1H, H-3′), 6.16 (ddt, *J* = 18.0, 11.5, 6.0 Hz, 1H, H-2′), 6.86 (s, 1H, H-4), 7.33 (td, *J* = 8.0, 1.0 Hz, 1H, H-7), 7.39 (dd, *J* = 8.0, 1.0 Hz, 1H, H-5), 7.65 (td, *J* = 8.5, 1.8 Hz, 2H, H-6), 8.27 (dd, *J* = 8.0, 1.8 Hz, 1H, H-8). ^13^C-NMR (125 MHz, CDCl_3_) δ: 30.8 (C-1′), 44.6 (C-3″), 50.4 (C-2″), 56.0 (O*C*H_3_), 74.1 (C-1″), 98.8 (C-4), 113.5 (C-9a), 114.5 (C-3′), 116.9 (C-5), 122.5 (C-8a), 123.6 (C-7), 126.7 (C-8), 133.8 (C-6), 135.6 (C-1), 137.5 (C-2′), 142.9 (C-2), 155.0 (C-4b), 155.71 (C-4a), 157.77 (C-3), 177.1 (*C*O-9). HRMS (EI^+^) calculated for C_20_H_18_O_5_: 338.1154. Found: 338.1158.

General procedure for preparing imidazole-substituted xanthones **12a–f**

Imidazoles **13a–c** (2.0 mol equiv.) were each added to a solution of the respective alkoxy-substituted xanthone **11a–b** (1.0 mol equiv.) in methanol (5 mL) at rt, and the resulting mixture was stirred at reflux for 24 h. After the reaction was completed (as monitored by TLC), the residual mixture was filtered, the solvent was removed under vacuum, and the residue was purified by flash chromatography over silica gel (CH_2_Cl_2_/MeOH, 98:02), generating the corresponding product. 

2-(2-Hydroxy-3-(1H-imidazol-1-yl)propoxy)-3-methoxy-9H-xanthen-9-one (**12a**). A white solid (63%), *Rf* 0.40 (CH_2_Cl_2_/MeOH, 9:1); mp 146–148 °C. ^1^H-NMR (600 MHz, DMSO-*d_6_*) δ: 3.90 (dd, *J =* 9.9, 4.8 Hz, 1H, H-1′), 3.94 (dd, *J =* 9.9, 5.4 Hz, 1H, H-1′), 3.96 (s, 3H, OC*H*_3_), 4.06 (dd, *J* = 13.8, 7.2 Hz, 1H, H-3′), 4.13 (sept, *J* = 5.4 Hz, 1H, H-2′), 4.19 (dd, *J =* 13.8, 3.6 Hz, 1H, H-3′), 5.58 (brs, 1H, OH), 6.87 (s, 1H, H-4″), 7.16 (s, 1H, H-5″), 7.21 (s, 1H, H-4), 7.43 (brt, *J* = 7.2 Hz, 1H, H-7), 7.49 (s, 1H, H-1), 7.56 (brd, *J* = 7.8 Hz, 1H, H-5), 7.59 (brs, 1H, H-2″), 7.80 (td, *J =* 8.4, 1.8 Hz, 1H, H-6), 8.15 (dd, *J* = 8.4, 1.8 Hz, 1H, H-8). ^13^C-NMR (150 MHz, DMSO-*d_6_*) δ: 49.3 (C-3′), 56.6 (O*C*H_3_), 68.2 (C-2′), 70.7 (C-1′), 100.6 (C-4), 106.5 (C-1), 113.9 (C-9a), 117.9 (C-5), 120.1 (C-5″), 120.8 (C-8a), 124.2 (C-7), 125.7 (C-8), 128.1 (C-4″), 134.7 (C-6), 135.2 (C-2″), 145.7 (C-2), 152.5 (C-4a), 155.9 (C-4b), 156.3 (C-3), 175.1 (*C*O-9). HRMS (EI^+^) calculated for C_20_H_18_N_2_O_5_: 366.1216. Found: 366.1206.

2-(2-Hydroxy-3-(2-phenyl-1H-imidazol-1-yl)propoxy)-3-methoxy-9H-xanthen-9-one (**12b**). A white solid (35%), *Rf* 0.54 (CH_2_Cl_2_/MeOH, 9:1); mp 157–159 °C. ^1^H-NMR (750 MHz, CDCl_3_) δ: 3.96 (s, 3H, OC*H*_3_), 3.97 (dd, *J =* 9.4, 6.0 Hz, 1H, H-1′), 4.02 (dd, *J =* 9.4, 4.5 Hz, 1H, H-1′), 4.22 (dd, *J* = 13.5, 7.5 Hz, 1H, H-3′), 4.28–4.31 (m, 1H, H-2′), 4.34 (dd, *J =* 13.5, 4.5 Hz, 1H, H-3′), 5.30 (brs, 1H, OH), 6.88 (s, 1H, H-4), 7.11 (s, 1H, H-4″), 7.19 (s, 1H, H-5″), 7.35–7.42 (m, 4H, H-7, H-3‴, H-4‴), 7.46 (d, *J* = 8.2 Hz, 1H, H-5), 7.57 (s, 1H, H-1), 7.60 (d, *J* = 6.7 Hz, 1H, H-2‴), 7.70 (td, *J =* 8.2, 1.5 Hz, 1H, H-6), 8.32 (dd, *J* = 8.2, 1.5 Hz, 1H, H-8). ^13^C-NMR (187.5 MHz, CDCl_3_) δ: 49.2 (C-3′), 56.2 (O*C*H_3_), 69.4 (C-2′), 70.5 (C-1′), 99.8 (C-4), 107.8 (C-1), 114.7 (C-9a), 117.6 (C-5), 121.43 (C-5″), 121.45 (C-8a), 123.8 (C-7), 126.5 (C-8), 128.52 (C-4″), 128.55 (C-3‴), 128.7 (C-4‴), 129.1 (C-2‴), 130.4 (C-1‴), 134.1 (C-6), 146.2 (C-2), 148.1 (C-2″), 152.8 (C-4a), 155.8 (C-3), 156.0 (C-4b), 175.9 (*C*O-9). HRMS (EI^+^) calculated for C_29_H_26_N_2_O_5_: 442.1529. Found: 442.1521.

2-(3-(2-(4-Chlorophenyl)-1H-imidazol-1-yl)-2-hydroxypropoxy)-3-methoxy-9H-xanthen-9-one (**12c**). A white solid (75%), *Rf* 0.76 (CH_2_Cl_2_/MeOH, 9:1); mp 235 °C (decomposition). ^1^H-NMR (750 MHz, DMSO-*d_6_*) δ: 3.89 (dd, *J* = 9.7, 5.2 Hz, 1H, H-1″), 3.92 (s, 3H, OC*H*_3_), 3.94 (dd, *J* = 9.7, 4.5 Hz, 1H, H-1″), 4.15–4.20 (m, 2H, H-2′, H-3′), 4.27–4.32 (m, 1H, H-3′), 5.62 (d, *J =* 5.2 Hz, 1H, OH), 7.03 (s, 1H, H-4″), 7.21 (s, 1H, H-4), 7.39 (s, 1H, H-5″), 7.42 (s, 1H, H-1), 7.42–7.48 (m, 3H, H-7, H-2‴), 7.62 (d, *J* = 8.2 Hz, 1H, H-5), 7.64–7.68 (m, 2H, H-3‴), 7.83 (td, *J* = 8.2, 1.5 Hz, 1H, H-6), 8.17 (dd, *J* = 7.5, 1.5 Hz, 1H, H-8). ^13^C-NMR (187.5 MHz, DMSO-*d_6_*) δ: 49.0 (C-3′), 56.4 (O*C*H_3_), 68.1 (C-2′), 70.0 (C-1′), 100.4 (C-4), 105.9 (C-1), 113.8 (C-9a), 117.8 (C-5), 120.7 (C-8a), 122.6 (C-5″), 124.1 (C-7), 125.7 (C-8), 127.9 (C-4″), 128.3 (C-3‴), 129.7 (C-1‴), 130.4 (C-2‴), 132.9 (C-4‴), 134.6 (C-6), 145.4 (C-2), 145.9 (C-2″), 152.0 (C-4a), 155.4 (C-4b), 155.6 (C-3), 174.5 (*C*O-9). HRMS (EI^+^) calculated for C_26_H_21_N_2_O_5_Cl: 476.1139. Found: 476.1137.

1-Allyl-2-(2-hydroxy-3-(1H-imidazol-1-yl)propoxy)-3-methoxy-9H-xanthen-9-one (**12d**). A white solid (78%), *Rf* 0.45 (CH_2_Cl_2_/MeOH, 9:1); mp 187–189 °C. ^1^H-NMR (500 MHz, DMSO-*d_6_*) δ: 3.78 (brd, *J* = 5.0 Hz, 2H, H-1″), 3.97 (s, 3H, OC*H*_3_), 4.01–4.12 (m, 2H, H-2″, H-3″), 4.17 (brd, *J* = 6.0 Hz, 2H, H-1′), 4.21–4.28 (m, 1H, H-3″), 4.90 (dd, *J* = 10.0, 2.0 Hz, 1H, H-3′), 4.95 (dc, *J* = 17.5, 2.0 Hz, 1H, H-3′), 5.43 (d, *J* = 5.5 Hz, 1H, OH), 6.01 (ddt, *J* = 16.5, 12.0, 5.5 Hz, 1H, H-2′), 6.93 (brs, 1H, H-4‴), 7.17 (s, 1H, H-4), 7.23 (brs, 1H, H-5‴), 7.43 (td, *J* = 8.0, 1.0 Hz, 1H, H-7), 7.56 (dd, *J* = 8.0, 0.5 Hz, 1H, H-5), 7.65 (brs, 1H, H-2‴), 7.80 (td, *J* = 8.5, 1.8 Hz, 1H, H-6), 8.13 (dd, *J* = 8.0, 1.8 Hz, 1H, H-8). ^13^C-NMR (125 MHz, CDCl_3_) δ: 30.2 (C-1′), 49.3 (C-3″), 56.4 (O*C*H_3_), 68.9 (C-2″), 74.4 (C-1″), 99.7 (C-4), 112.3 (C-9a), 114.8 (C-3′), 117.2 (C-5), 120.4 (C-5‴), 121.7 (C-8a), 124.0 (C-7), 126.0 (C-8), 128.3 (C-4‴), 134.1 (C-1), 134.5 (C-6), 137.6 (C-2′), 138.3 (C-2‴), 142.6 (C-2), 154.4 (C-4b), 155.0 (C-4a), 157.8 (C-3), 176.0 (*C*O-9). HRMS (EI^+^) calculated for C_23_H_22_N_2_O_5_: 406.1529. Found: 406.1527.

1-Allyl-2-(2-hydroxy-3-(2-phenyl-1H-imidazol-1-yl)propoxy)-3-methoxy-9H-xanthen-9-one (**12e**). A white solid (38%), *Rf* 0.54 (CH_2_Cl_2_/MeOH, 9:1); mp 187–189 °C. ^1^H-NMR (750 MHz, DMSO-*d_6_*) δ: 3.77 (dd, *J* = 9.0, 4.9 Hz, 1H, H-1″), 3.85 (dd, *J* = 9.0, 4.9 Hz, 1H, H-1″), 3.92 (s, 3H, OC*H*_3_), 4.07–4.17 (m, 4H, H-1′, H-2″, H-3″), 4.33 (d, *J* = 10.5 Hz, 1H, H-3″), 4.84–4.90 (m, 2H, H-3′), 5.55 (brs, 1H, OH), 5.95 (ddt, *J* = 16.5, 10.5, 6.0 Hz, 1H, H-2′), 7.03 (d, *J* = 0.7 Hz, 1H, H-4‴), 7.14 (s, 1H, H-4), 7.38–7.43 (m, 3H, H-7, H-5‴, H-4^IV^), 7.43–7.48 (m, 2H, H-3^IV^), 7.54 (d, *J* = 8.2 Hz, 1H, H-5), 7.68–7.15 (m, 2H, H-2^IV^), 7.79 (td, *J* = 8.2, 1.5 Hz, 1H, H-6), 8.11 (dd, *J* = 7.5, 1.5 Hz, 1H, H-8). ^13^C-NMR (187.5 MHz, DMSO-*d_6_*) δ: 30.1 (C-1′), 49.3 (C-3″), 56.4 (O*C*H_3_), 69.0 (C-2″), 74.7 (C-1″), 99.6 (C-4), 112.3 (C-9a), 114.7 (C-3′), 117.2 (C-5), 121.7 (C-8a), 122.1 (C-5‴), 124.0 (C-7), 126.0 (C-8), 127.6 (C-4‴), 128.2 (C-4^IV^), 128.3 (C-3^IV^), 128.8 (C-2^IV^), 131.1 (C-1^IV^), 134.1 (C-1), 134.5 (C-6), 137.6 (C-2′), 142.5 (C-2), 147.0 (C-2‴), 154.4 (C-4b), 155.0 (C-4a), 157.7 (C-3), 176.0 (*C*O-9). HRMS (EI^+^) calculated for C_29_H_26_N_2_O_5_: 482.1842. Found: 482.1838.

1-Allyl-2-(3-(2-(4-chlorophenyl)-1H-imidazol-1-yl)-2-hydroxypropoxy)-3-methoxy-9H-xanthen-9-one (**12f**). A white solid (81%), *Rf* 0.52 (CH_2_Cl_2_/MeOH, 9:1); mp 116–118 °C. ^1^H-NMR (750 MHz, DMSO-*d_6_*) δ: 3.78 (dd, *J* = 9.7, 4.5 Hz, 1H, H-1″), 3.85 (dd, *J* = 9.7, 4.5 Hz, 1H, H-1″), 3.92 (s, 3H, OC*H*_3_), 4.09 (dd, *J* = 13.5, 6.0 Hz, 1H, H-3″), 4.10–4.16 (m, 3H, H-1′, H-2″), 4.33 (dd, *J* = 18.0, 7.5 Hz, 1H, H-3″), 4.84–4.90 (m, 2H, H-3′), 5.56 (brs, 1H, OH), 5.95 (ddt, *J* = 17.2, 10.5, 6.0 Hz, 1H, H-2′), 7.05 (s, 1H, H-4‴), 7.15 (s, 1H, H-4), 7.40–7.45 (m, 2H, H-7, H-5‴), 7.48–7.52 (m, 2H, H-3^IV^), 7.54 (d, *J* = 8.2 Hz, 1H, H-5), 7.72–7.76 (m, 2H, H-2^IV^), 7.79 (td, *J* = 8.2, 1.5 Hz, 1H, H-6), 8.11 (dd, *J* = 8.2, 1.5 Hz, 1H, H-8). ^13^C-NMR (187.5 MHz, DMSO-*d_6_*) δ: 30.1 (C-1′), 49.4 (C-3″), 56.4 (O*C*H_3_), 69.0 (C-2″), 74.7 (C-1″), 99.7 (C-4), 112.3 (C-9a), 114.6 (C-3′), 117.2 (C-5), 121.7 (C-8a), 122.5 (C-5‴), 124.0 (C-7), 126.0 (C-8), 127.9 (C-4‴), 128.4 (C-3^IV^), 129.9 (C-1^IV^), 130.5 (C-2^IV^), 133.1 (C-4^IV^), 134.0 (C-1), 134.5 (C-6), 137.6 (C-2′), 142.5 (C-2), 145.9 (C-2‴), 154.4 (C-4b), 155.0 (C-4a), 157.7 (C-3), 176.0 (*C*O-9). HRMS (EI^+^) calculated for C_29_H_25_ClN_2_O_5_: 516.1452. Found: 516.1435.

### 3.3. Biological Evaluation

#### 3.3.1. α-Glucosidase Inhibition Assay

The inhibitory activity of the compounds on α-glucosidase inhibition was quantified according to the method described by Salehi et al., with slight modifications [64]. A reaction was prepared by mixing 20 µL α-glucosidase solution (0.5 unit/mL), 120 µL 0.1 M phosphate buffer (pH 6.9), and 10 µL of the samples at concentrations from 400 µM to 4.0 µM. The solution was incubated in a 96-well microplate at 37 °C for 15 min. The enzymatic reaction was initiated by adding 20 µL of 5 mM *p*-NPG solution to 0.1 M phosphate buffer (pH 6.9), followed by incubation at 37 °C for 15 min. The reaction was stopped by adding 80 µL of 0.2 M sodium carbonate solution, and absorbance was read at 405 nm in a microplate reader (Epoch^®^, BioTek Instrument, Winooski, EUA). The reaction system without any test compounds was used as control, whereas the system without α-glucosidase served as blank for correcting background absorbance. The rate of α-glucosidase inhibition exerted by each sample was calculated with Equation (1):(1)$$\%\;inhibition=\left(\frac{Control\;absorbance-sample\;absorbance}{Control\;absorbance}\right)\times 100$$

All measurements were performed in quadruplicate and the values are expressed as the mean ± standard deviation. 

#### 3.3.2. α-Amylase Inhibition Assay

α-Amylase inhibitory activity was quantified according to the method developed by Chokki et al., with some modifications [65]. The reaction mixture consisting of 50 µL of 0.1 M phosphate buffer (pH 6.8), 10 µL of α-amylase solution (5.0 unit/mL), and 20 µL of the sample at various concentrations (from 100 µM to 5.0 µM) was placed in a 96-well plate and pre-incubated at 37 °C for 15 min, and 20 µL of 1% soluble starch (0.1 M phosphate buffer, pH 6.8) was then added as a substrate and incubated at 37 °C for 45 min. Finally, 100 µL of 3,5-dinitrosalicylic acid (DNS) was added and heated at 100 °C for 20 min, and absorbance was then read at 540 nm in a microplate reader (Epoch, BioTek^®^). The reaction system without any test compound was used as the control, and the system without α-amylase served as a blank for correcting background absorbance. The percentage of α-amylase inhibition was calculated for each sample with Equation (2):(2)$$\%\;inhibition=\left(\frac{Control\;absorbance-sample\;absorbance}{Control\;absorbance}\right)\times 100$$

All measurements were performed in quadruplicate and the values are expressed as the mean ± standard deviation. 

#### 3.3.3. Kinetic Study

To explore the type of enzyme inhibition, kinetic studies were carried out with α-glucosidase and α-amylase using a methodology such as that described in the inhibitory activity assays. The alkoxy-substituted xanthones were evaluated at four different concentrations according to their IC_50_. Various concentrations of substrates were used for each of the enzymes in the range of 0.5–5.0 mM for *p*-NPG in α-glucosidase and 0.1–1.0% for α-amylase. The type of inhibition for each test compound was determined by utilizing double reciprocal plots. Inhibition constants (K_I_) were calculated from substrate versus reaction rate curves using nonlinear regression of the enzyme inhibition kinetic function [55,65]. 

### 3.4. DPPH Radical Scavenging Assay

The scavenging of free radicals by the synthesized compounds was assessed based on the previously reported DPPH radical assay, with slight modifications [66]. The reaction mixture consisted of 50 µL of compound in DMSO at various concentrations (from 2.5 mM to 0.2 mM) and 150 µL of DPPH solution at 133.33 µM in absolute ethanol. The reaction components were added at a ratio of 1:3 (*v*/*v*). The mixture was incubated at 37 °C for 30 min before absorbance was read at 517 nm using a microplate reader (Epoch, BioTek^®^). Butylhydroxytoulene (BHT) served as the positive control. Scavenging capacity (%) is expressed as the percentage decrease in DPPH.
SC% = [(A_control_ − A_test_)/A_control_] × 100
where A_control_ is absorbance of the DPPH solution (control) and A_test_ is the absorbance of the solution of DPPH and one of the compounds.

### 3.5. Docking Studies

The molecular docking studies were carried out in the AutoDock 4 program [67] using the crystallized proteins of isomaltase from *Saccharomyces cerevisiae* (PDB: 3A4A) and human pancreatic α-amylase (PDB: 1B2Y) in complex with the inhibitor acarbose. In these proteins, water molecules were removed, hydrogen atoms were added to the polar atoms (considering pH at 7.4), and Kollman charges were assigned with AutoDock Tools 1.5.6. The 3D structures of acarbose (**14**), **1** (natural xanthone), alkoxy-xanthones **6c**, **6e**, **9a**, and **9b**, and imidazole-xanthones **10c** and **10f** were sketched in two dimensions (2D) with ChemSketch and then converted to 3D in a mol2 format using the Open Babel GUI program [68]. The ligands were optimized with PM6 on Gaussian 98 software to obtain the lowest energy conformation. All the possible rotatable bonds, torsion angles, atomic partial charges, and non-polar hydrogens were determined for each ligand. In AutoDockTools, the grid dimensions for α-glucosidase were 78 × 60 × 78 Å^3^ with points separated by 0.375 Å and centered at X = 26.313, Y = −3.544, and Z = 26.146. The grid dimensions for α-amylase were 90 × 70 × 66 Å^3^ with points separated by 0.375 Å and centered at X = 16.758, Y = 8.692, and Z = 49.959. The hybrid Lamarckian genetic algorithm was applied for minimization and utilized default parameters. A total of one hundred docking runs were conducted to determine the conformation with the lowest binding energy (kcal/mol), which was adopted for all further simulations. AutoDockTools was used to prepare the script and files as well as to visualize the docking results, and these were edited with the Discovery 4.0 client.

### 3.6. Physicochemical Properties

The physicochemical properties of compounds **1**, **6a**, **6c**, **6e**, **7**, **9a–b**, **10c**, **10f**, **11b**, and **12b–f** and acarbose (**14**) were generated in silico with OSIRIS DataWarrior V4.7.2 (http://www.organic-chemistry.org/prog/peo/ accessed on 15 January 2023) [59]. Druglikeness was evaluated based on Lipinski’s rule of five [69].

## 4. Conclusions

Alkoxy- and imidazole-substituted xanthones **6–12** were synthesized and their inhibitory activity on α-glucosidase and α-amylase enzymes was evaluated. Compared to the reference drug acarbose (**14**), the inhibitory activity of **6c**, **6e**, and **9b** was higher for α-glucosidase and lower for α-amylase, reflecting a desirable outcome. Based on structure–activity analysis of the results, a 4-bromobutoxy or 4′-chlorophenylacetophenone moiety in the molecule favors greater inhibition of α-glucosidase versus α-amylase. In contrast, inserting a 2-(4-chlorophenyl)butoxyimidazole moiety (**10c**) produces lower α-glucosidase inhibition and higher α-amylase inhibition. The mechanism of the enzymatic inhibition of **6c**, **10c**, and **9b** was determined, establishing that for α-glucosidase they are mixed inhibitors, while for α-amylase, **6c** is a competitive inhibitor and **10c** is mixed. The docking studies revealed that the π-stacking and hydrophobic effects of the aromatic moiety at the C-2 position of the xanthone backbone play a key role in the interaction with the active sites of both α-glucosidase and α-amylase. Additionally, drug prediction and ADMET studies suggest that compounds **6c**, **6e**, and **9b** are candidates for the development of new selective α-glucosidase inhibitors with antidiabetic potential.

## Data Availability

The data presented in this study are available in the Appendix A.

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
