# Peer review of "Synthesis and Molecular Docking Studies of Alkoxy- and Imidazole-Substituted Xanthones as α-Amylase and α-Glucosidase Inhibitors"

_molecules, 2023, doi:10.3390/molecules28104180_

Round 1
Reviewer 1 Report
The manuscript reports the synthesis and molecular docking studies of alkoxy- and imid-2 azole-substituted xanthones and their evaluation as α-amylase and α-glucosidase inhibitors.
A good experimental design was applied, all the details for the synthesis and structural characterization of the compounds is described, recent and relevant literature is presented.
There is, however, some points that need clarification. The most important concerns the main objective of the work and the importance of the synthesized compounds. In the abstract, the authors write “Current antidiabetic drugs have severe side effects, which may be minimized by new selective molecules that strongly inhibit α-glucosidase and weakly inhibit α-amylase”. If so, why the authors claim in the conclusions that compounds 6c and 6e are good candidates for the development of new selective α-glucosidase inhibitors with antidiabetic potential? Shouldn’t it be compound 9b? It’s the strongest alpha-glucosidase inhibitor. Moreover, according to the enzymatic studies, this compound is the most active for alpha-glucosidase and was no influence on alpha-amylase activity. 9b should be the lead compound.
Why the authors, in the enzymatic mechanism of inhibition, only studied compounds 6a and 10c? Shouldn´t it be compound 9b? Or 6c and 6e? At least 3 concentrations of the inhibitors should be used.
Why the authors in the docking studies, used compounds 6c, 6e, 9b for alpha-glucosidase and compounds 1, 10c and 10f for alpha-amylase? Why are they always changing the target compounds? There is no coherence about the importance of each compound…
Other comments:
In the discussion part, the authors should present the real importance of the substituents. According to the results, the introduction of the imidazolyl group had no effect on the activity since the starting materials and the products have almost the same IC50... and they are ineffective inhibitors...Generally, compounds 10-12 are less effective than the corresponding starting materials…
In the enzymatic inhibitory assays, the authors should test all compounds for alpha-amylase activity.
Present the 1H and 13C NMR main assignments of the target compounds used to confirm the chemical structures.
Based on these evidences, the manuscript needs a careful revision to be accepted for publication.
Author Response
Please see the attachment. Very thank you.

Reviewer 2 Report
1. Why did authors choose acarbose to make a comparison as we have many new antidiabetic drugs in the market, e.g. metformin, miglitol, etc.?
2. Table 1. Authors should identify the significant differences.
3. Please show your DPPH graph to prove your IC50 value because the authors only report the %DPPH. The authors should mention the concentration of compounds tested in Table 1.
4. Line 863. "....samples at different concentrations." Please state the range. The same goes for Line 882.
5. Section 3.4. What concentration of samples was tested in this assay? Please mention it.
-
Author Response

(The authors gave the same response as above.)

Reviewer 3 Report
Comments and suggestions for authors:
This research supplied some useful and novel scientific information having valuable reference to new drug investigation. The background of this study is good, in addition, it has a suitable designation and good presentation. However, some major points that should be noticed and revised for further consideration:
- The original novelty of your work should carefully indicate clearly in the abstract, results, and conclusion sections.
- Though the Prediction of drug-like properties and ADMET were conducted, no information was mentioned in the abstract as well as in the Conclusions sections. Please add.
- Please showed the data (NMR spectra, Mass, ....) in the supplementary section.
- If the synthesis compounds are impurity, the activity cannot be accepted. How do you know all the compounds have good purity grades?? Please add this information and the HPLC profiles of each compound.
- Can the author know if the compounds were interacting with enzymes at the catalytic site or binding sites? Please add the data and 3D structures of the binding sites of ligands and the catalytic site of the enzyme.
Moderate editing of English language is requied
Author Response

(The authors gave the same response as above.)

Reviewer 4 Report
This manuscript reports a study on “Synthesis and molecular docking studies of alkoxy- and imidazole-substituted xanthones as α-amylase and α-glucosidase inhibitors”. I recommend following changes before manuscript can be considered for publication.
1. The author should re-arrange and re-write the abstract in the following order (Problem of research, aim of study, remarkable methodology, remarkable results, and significance of study).
2. The images used in manuscript are unclear and blurred.
3. References need to be unified.
4. Please find the particle size of synthesized materials.
5. Research article still lacking with recent literature authors should read below recent relevant reference and consider for citation
doi.org/10.1021/acsomega.2c05133, Materials Science and Engineering: B 272, 115365,2021
English should be improved
Author Response

(The authors gave the same response as above.)

Round 2
Reviewer 1 Report
The authors made an effort to revise and include the suggestions given by the referee.
It can be accepted for publication.
Reviewer 2 Report
accept in present form
accept in present form
Reviewer 3 Report
The manuscript was significantly revised accordingly to the reviewer's comments.
Decision: Accepted
Minor editing of English language required